# Fish Detection and Classification for Automatic Sorting System with an Optimized YOLO Algorithm

Ari Kuswantori [1], Taweepol Suesut [1,*], Worapong Tangsrirat [1], Gerhard Schleining [2] and Navaphattra Nunak [3]

[1] Department of Instrumentation and Control Engineering, School of Engineering, King Mongkut's Institute of Technology Ladkrabang (KMITL), Bangkok 10520, Thailand; 63601247@kmitl.ac.th (A.K.); worapong.ta@kmitl.ac.th (W.T.)

[2] Department of Food Science and Technology, University of Natural Resources and Life Sciences Vienna (BOKU), 1190 Vienna, Austria; gerhard.schleining@boku.ac.at

[3] Department of Food Engineering, School of Engineering, King Mongkut's Institute of Technology Ladkrabang (KMITL), Bangkok 10520, Thailand; navaphattra.nu@kmitl.ac.th

\* Correspondence: taweepol.su@kmitl.ac.th

**Featured Application: In the future, the application of this study is very feasible and very close to being implemented for the auto-sorting system for various fish or other objects, in the fish industry or other industries, with deep learning and machine vision technology.**

**Abstract:** Automatic fish recognition using deep learning and computer or machine vision is a key part of making the fish industry more productive through automation. An automatic sorting system will help to tackle the challenges of increasing food demand and the threat of food scarcity in the future due to the continuing growth of the world population and the impact of global warming and climate change. As far as the authors know, there has been no published work so far to detect and classify moving fish for the fish culture industry, especially for automatic sorting purposes based on the fish species using deep learning and machine vision. This paper proposes an approach based on the recognition algorithm YOLOv4, optimized with a unique labeling technique. The proposed method was tested with videos of real fish running on a conveyor, which were put randomly in position and order at a speed of 505.08 m/h and could obtain an accuracy of 98.15%. This study with a simple but effective method is expected to be a guide for automatically detecting, classifying, and sorting fish.

**Keywords:** automatic fish sorting; fish classification; fish recognition; YOLO; computer and machine vision

## 1. Introduction

Automatic fish detection, recognition, and classification are popular and intriguing areas of research. Numerous researchers are engaged in its development for underwater and out-of-water environments [1–6]. Recognizing fish in underwater conditions, especially in deep ocean environments, is advantageous for fish population control and sustainability [3,7–11] and supports the development of the IoUT (Internet of Underwater Things) [12,13]. Recognizing fish in settings other than water, especially for farming fish, is useful in aquaculture, for example, for automatic sorting processes, fish quality monitoring, and other activities [4–6,14–19].

Automatic fish recognition using machine and computer vision is a key part of automating the fish industry, which is part of the food industry, to boost productivity in aquaculture. It deals with the problems of high food demand and the possibility that there won't be enough food in the future because of the growing world population and

the effects of global warming and climate change [20,21]. One of them is for the automatic sorting system, which has already been mentioned and is being worked on by many researchers [22–25]. Automatic detection and classification of moving fish is the key to automatic sorting systems in the fish culture industry, and it has some unique challenges. The main challenge is the speed, and others are the random position of the fish, the background, the deformed condition of the fish, and the fact that fish usually have a similar visual appearance [14].

For moving fish recognition, many studies have been carried out. The study in [7] employed CNN (Convolutional Neural Network) to classify fish by training them with the number of species and their environments, such as reef bottoms and water. They applied their proposed method to 116 underwater fish videos recorded using a GoPro underwater camera. The best results were achieved in classifying 9 of the 20 types of fish that appear most often in the videos. In [8], a multi-cascade object detection network with an ensemble of seven CNN components and two RPNs (Region Proposal Network) linked by sequentially jointly trained LSTMs (Long Short-Term Memory units) was performed. For training and testing, they used a set of 18 underwater fish videos that were also recorded with a GoPro underwater camera. Even though their proposed method can reliably find and count fish objects in a variety of benthic backgrounds and lighting conditions, it is only used to find fish and not to classify them. Using classic CNNs like these also has advantages when applied to other sectors, such as agriculture [26], or in other broad cases, such as detecting fine scratches [27]. The moving fish recognition in [9] utilized Optical flow, GMM (Gaussian Mixture Models), and ResNet-50, then combined the output with YOLOv3. The combination of those methods enabled the robust detection and classification of fish, which was applied to the LifeCLEF 2015 benchmark dataset from the Fish4Knowledge repository [28] and a dataset collected by the University of Western Australia (UWA) which was explained in detail by [29]. The GMM and Pixel-wise posteriors were proposed in [11], and then further developed by combining them with CNN [30]. They also used a fish dataset extracted from the Fish4Knowledge repository in their work. Similar to the work [8], the approaches proposed in their papers were only for detecting fish without classifying them.

Abinaya et al. [14] segmented the fish into three parts; head, body, and scales. Then an Alex-Net was used to classify each of these parts. Naive Bayesian Fusion (NBF) was then utilized to determine the final classification results. The accuracy obtained from this approach was quite well applied to the Fish-Pak [31] and BYU (Bringham Young University) [32] datasets. Still, the fish images used were only static, even though the narrative of this work was intended for an automatic sorting system. Mohamed et al. [16] proposed an approach for fish detection in aquaculture ponds. Image enhancement was used to improve fish detection in cloudy water conditions, and then YOLOv3 was utilized to detect the fish. However, this approach is not intended for classification but for counting and tracking fish trajectory. Xu et al. [17] applied Faster R-CNN and compared it with YOLOv3 to detect and record fish trajectory to study its behavior and relationship to ammonia levels in pond water.

However, the works reported in [7–9,11,30] recognized moving fish for underwater (ocean) environments, while some only detected fish without classifying them. Moving fish in aquaculture was discovered by [16,17], but not used for classification. The work in [14] classified fish with narration for the automatic sorting system, but the datasets used were static images. To the best of the authors' knowledge, there is no public dataset for cultured fish that run on conveyors, and there is no published work to detect and classify moving fish for the fish culture industry, especially for automatic sorting based on fish species using deep learning and computer vision. This paper will fill that gap, and it proposes a method for detecting and classifying fish and tests it on real videos of aquacultured freshwater fish moving along a conveyor belt for automatic sorting using deep learning and computer vision.

In summary, this work thus creates the following significant contributions:

1.  We compiled our own dataset of eight cultivated fish species. The dataset contains not only static images, but also videos of fish running randomly at two different speeds on a conveyor belt (low and high).
2.  This work employed YOLOv4, a very popular recognition algorithm, which was optimized with a unique labeling technique.
3.  A trial study with several schemes was also conducted to determine their effectiveness. Among them are schemes for using data for training, versions of YOLOv4, and comparisons of labeling techniques.

By using the proposed method and using real videos of freshwater fish running on a conveyor, it is anticipated that this work will serve as a guide and provide solutions to the challenges of detecting and classifying fish for automatic sorting, which is very close to the actual condition.

## 2. Materials and Methods

### 2.1. Images Dataset and the Experimental Set-Up

The purpose of this work is to develop an approach to automatically detect and classify fish for automatic sorting systems in the fish industry. As far as the authors know, there is no publicly available dataset for cultured fish run on conveyors that can be used for this purpose. For that, we created our own dataset for this work. We took fish samples from eight types of farmed fish species called "Ben-Cak", which are generally bred, sold, and consumed in and around Thailand. Three of these fish species are endemic, originally from the Mekong and Chao Phraya rivers, which are also in Thailand. The eight fish species are:

1.  Yeesok (*Labeo rohita*),
2.  Nuanchan (*Cirrhinus microlepsis*),
3.  Tapian (*Barbonymus gonionotus*),
4.  Nai (*Cyprinus carpio*),
5.  Jeen Ban (*Hypophthalmichthys molitrix*),
6.  Jeen To (*Hypophthalmichthys nobilis*),
7.  Nin (*Oreochromis niloticus*), and
8.  Sawai (*Pangasianodon hypophthalmus*).

Sample pictures of each type of fish can be seen in Figure 1. In Thailand, these fish are bred together (mixed) in the same pond on the fish farm, and then when harvested, these fish will be brought to the fish hub for sorting and weighing for further sale to consumers. For this reason, these fish are considered very suitable for this work because the sorting process carried out at the fish hub is still done manually by humans.

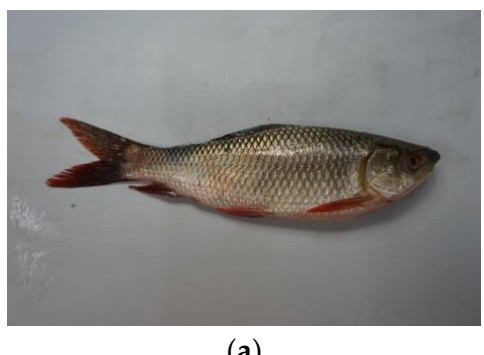 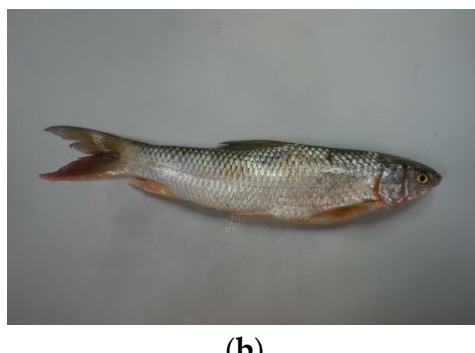

(**a**)        (**b**)

**Figure 1.** *Cont.*

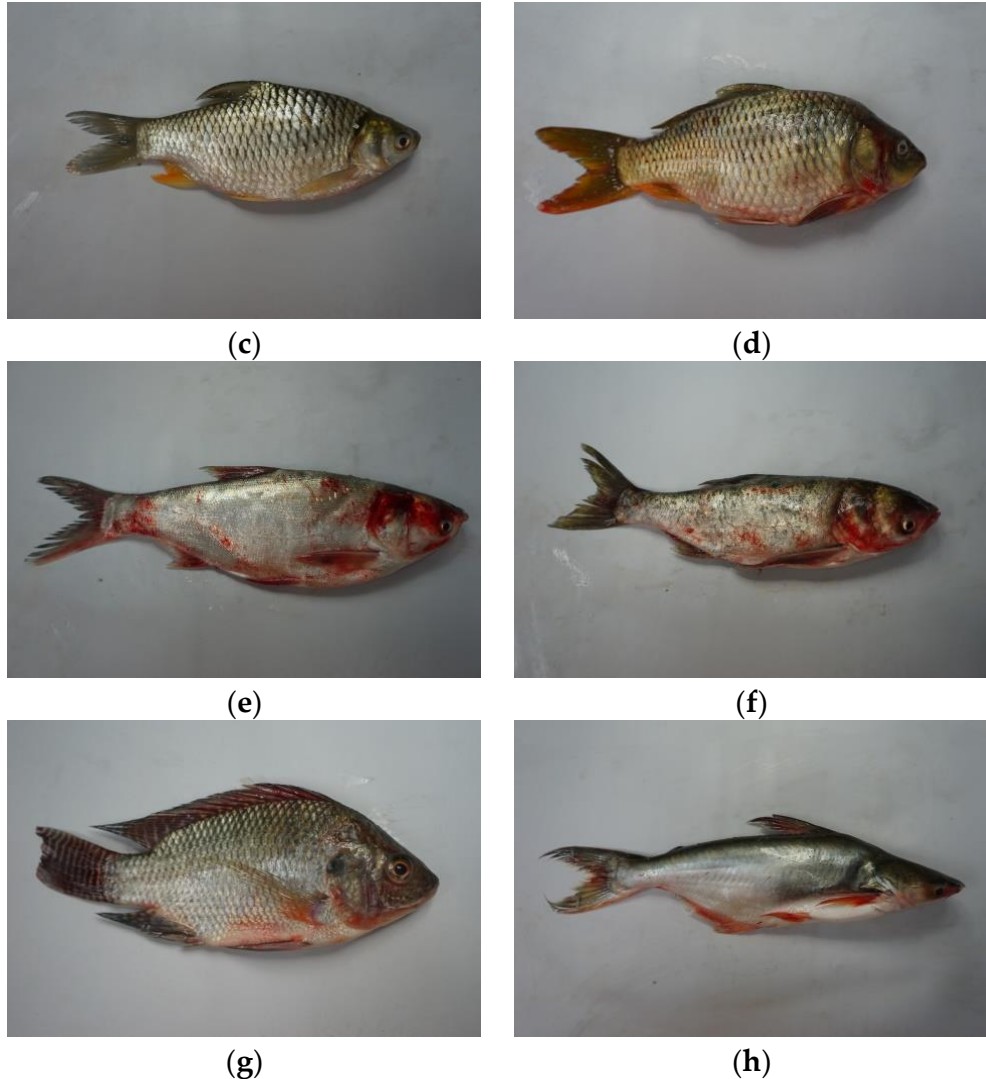

**Figure 1.** Sample picture of fish: (**a**) Yeesok; (**b**) Nuanchan; (**c**) Tapian; (**d**) Nai; (**e**) Jeen Ban; (**f**) Jeen To; (**g**) Nin; and (**h**) Sawai.

In creating the dataset, we did not just take static photos of each fish sample from several views, as shown in Figure 1. We ran a conveyor and randomly put the fish on it in both position and order. Then we recorded it with the overhead camera with two-speed settings; low (116.65 m/h) and high (505.08 m/h). The recordings were carried out in the Food Engineering laboratory at King Mongkut's Institute of Technology Ladkrabang (KMITL), Bangkok, Thailand, with room lighting conditions and additional light from an LED lamp. The measured light intensity was 846 lux/79 FC at all times during the recordings. We produced three videos; one low-speed video with a duration of 17 min and 13 s, and two high-speed videos, with a duration of 8 min and 24 s (later referred to as high-speed video 1) and 17 min and 13 s (later referred to as high-speed video 2). The camera used was a SONY Model ILCE-7 with a frame size setting of 1920 × 1080 (W × H) and a 29.97 frames/second frame rate.

In the videos produced, the background of the fish object is not entirely a conveyor, with a monotone condition. The conveyor is small and does not dominate the entire frame; it also has other objects in the background of the frame. So its condition makes this dataset more challenging. According to the authors, this is the first work that utilizes videos of aquacultured fish running on a conveyor. The experimental setup to create the dataset and the sample for capturing results can be seen in Figure 2. Tables 1 and 2 show images from the dataset that was created.

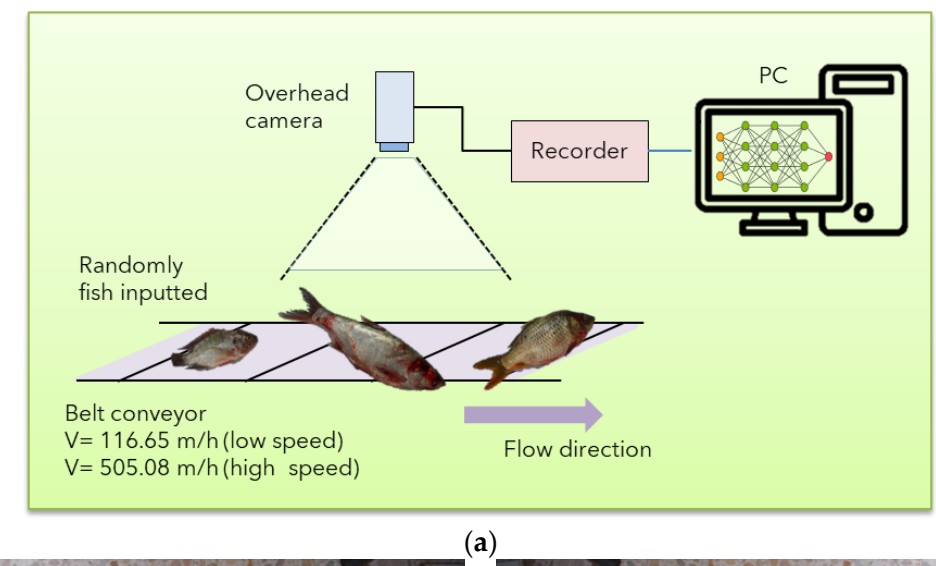

(a)

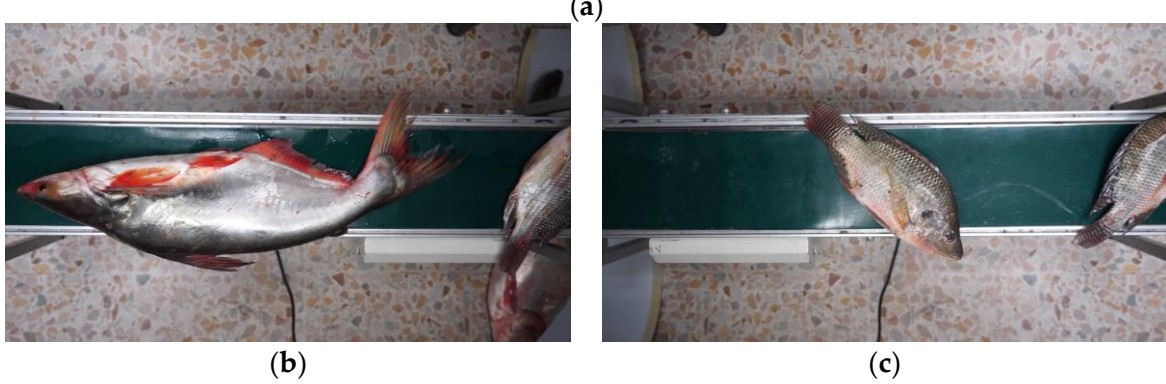

(b)                                     (c)

**Figure 2.** The experimental set-up for: (**a**) taking the videos; while (**b**,**c**) are examples of capturing video results.

**Table 1.** Images dataset (static pictures).

| Fish Class | Yeesok | Nuanchan | Tapian | Nai | Jeen Ban | Jeen To | Nin | Sawai |
|---|---|---|---|---|---|---|---|---|
| No. of images | 20 | 20 | 20 | 20 | 20 | 20 | 20 | 20 |
| Total | | | | | 160 | | | |
| Average per class | | | | | 20 | | | |

**Table 2.** Images dataset (videos).

| No. | Name | Conveyor Speed (m/h) | Duration | Note |
|---|---|---|---|---|
| 1 | low-speed video | 116.65 | 17 min 13 s | later extracted for training data (scheme 2) |
| 2 | high-speed video 1 | 505.08 | 8 min 24 s | for testing data |
| 3 | high-speed video 2 | 505.08 | 17 min 13 s | for testing data |

*2.2. Training Images and Augmentation*

       The algorithm for detecting and classifying (recognition) needs to be trained, so images for training should be prepared. This work uses two schemes for using or generating training images. First, using static pictures from each fish class, and second, generating and using extracted pictures taken from one of the video recordings from the dataset. Each image prepared is then augmented in both the first and second schemas to enrich the data for a better training process [14,33,34]. The augmentation techniques used in this work

are vertical and horizontal flips, which are suitable for the case of fish objects running on conveyors [35].

Scheme 1 employed 160 static images of each fish from eight classes. The images were then augmented to enrich the data. Each augmentation technique was applied to each original image. So, every one of the 160 new images was obtained using vertical and horizontal flip techniques. All images are then combined (original and augmented) and used as training images. In scheme 2, the low-speed video was used and extracted into 188 static images. Each fish that appears in the video was extracted by taking screenshots at three positions, as shown in Figure 3: when the fish appears in its entirety (a), at the exact center position (b), and just before the fish's snout or tail hits the right-hand frame border to leave the frame (c). Then augmentation was applied to enrich the data with the same method as in scheme 1. Furthermore, training images were also obtained by combining the original images with augmentations, as in scheme 1 as well. The training images and augmentations used in this work are summarized in Table 3.

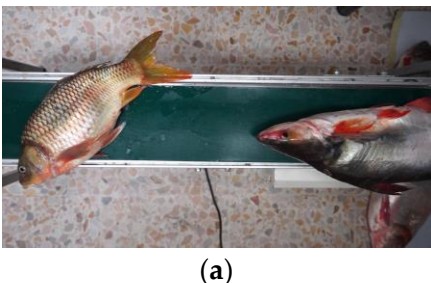 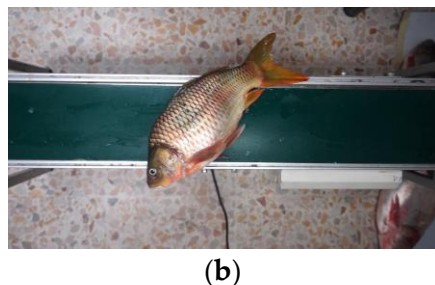 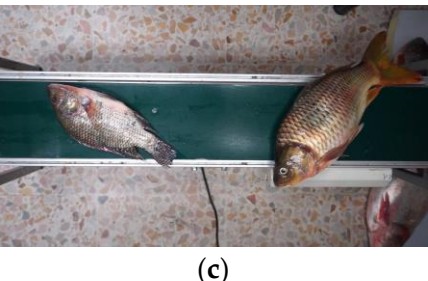

|(**a**)|(**b**)|(**c**)|

**Figure 3.** Example of extracting images for one fish from the low-speed video: (**a**) when the fish appears in its entirety; (**b**) at the exact center position; and (**c**) just before the fish's snout or tail hits the right-hand frame border to leave the frame.

**Table 3.** Images for training and augmentation.

| Scheme | Original Images (8 Classes) | Averaged per Class | Augmented Images | | | Total Images for Training | Averaged per Class |
|---|---|---|---|---|---|---|---|
| | | | Flip-Vertically | Flip-Horizontally | Total | | |
| Scheme-1 (from static pictures) | 160 | 20 | 160 | 160 | 320 | 480 | 60 |
| Scheme-2 (from extracted pictures) | 188 | 24 | 188 | 188 | 376 | 564 | 71 |

### 2.3. Labeling Techniques

The labeling process is utilized during the training step. There are three types of labeling used in this work: conventional, using landmarking, and a combination of conventional and landmarking. The conventional labeling technique is the most commonly used, but it has a weakness. The object's background is included so that the algorithm's feature extraction is carried out both on the object and background. It can make the learning process less effective and impact the recognition results [14,36]. Labeling using the landmarking technique was introduced by the authors for the first time in previous works [37,38]. Table 4 provides a summary of the previously stated works. This table contains important findings regarding the effect of certain labeling techniques on the recognition accuracy of fish. The landmarking technique can optimize the recognition algorithm significantly, especially in object recognition in various background conditions. Because by using this technique, all of the object's backgrounds can be removed so that the recognition algorithm will extract only the object's features without the background. The difference between conventional labeling techniques and using landmarking can be observed in Figure 4. For the third method,

the combination labeling technique is proposed in this work. This technique combines conventional imaging with landmarking, which will be explained in detail in Section 2.4.

**Table 4.** Summary of recent works by authors that are related to this work.

| References | Fish Object | Important Findings Related to This Work |
|:---:|:---|:---|
| [37] | - Static pictures. 6 classes.<br>- Constant backgrounds.<br>- Aquaculture fish with similar appearance and structural deformed.<br>- Taken from the Fish-Pak dataset. | Applying YOLOv4 with conventional labeling resulted in 14.29% higher accuracy than using landmarking for 6 classes. |
| [38] | - Static pictures. 4 classes.<br>- Various backgrounds.<br>- Ocean fish images captured in various background conditions, such as rocks, water, seaweed, etc.<br>- Taken from the BYU dataset. | Combining YOLOv4 with landmarking labeling techniques resulted in 4.94% higher accuracy than using conventional. |

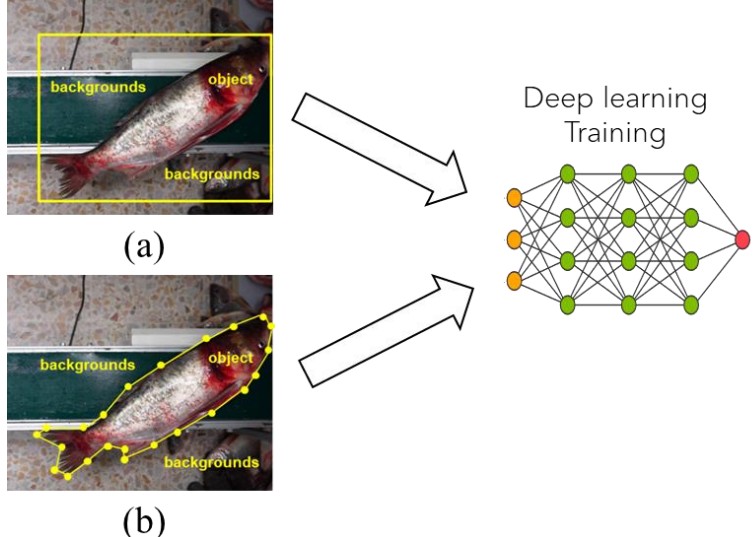

**Figure 4.** Labeling techniques in training process: (**a**) conventional; and (**b**) using landmarking.

### 2.4. YOLOv4, YOLOv4-Tiny, and the Training Process

In this work, YOLO (You Only Look Once) is employed as the recognition algorithm. This algorithm is very popular and known as a real-time object detector because of its speed and accuracy [39–43]. In addition, the version used is the relatively new, YOLO version 4, which was released in April 2020. We use this version because it accommodates our resources (currently, our resources only support this version), and we suppose that this version will achieve good performance with proper optimization. YOLOv4 consists of CSPDarknet53 as the backbone, SPP (Spatial Pyramid Pooling layer) & PAN (Path Aggregation Network) as the neck, and YOLOv3 as the head. A simple architecture of YOLOv4 is shown in Figure 5, and the output of this algorithm can be represented as [44]:

$$y = (P_c, B_y, B_x, B_w, B_h, C_1, C_2, \dots C_8) \tag{1}$$

where $y$ is the output of the YOLO, $Pc$ will become 1 if the algorithm detects objects (fish) and 0 if otherwise, $(B_y, B_x)$ is the center point of the produced bounding boxes of the fish, $(B_w, B_h)$ is the width and height of the bounding boxes, and $C_1$ untill $C_8$ represent each class for the fish [44].

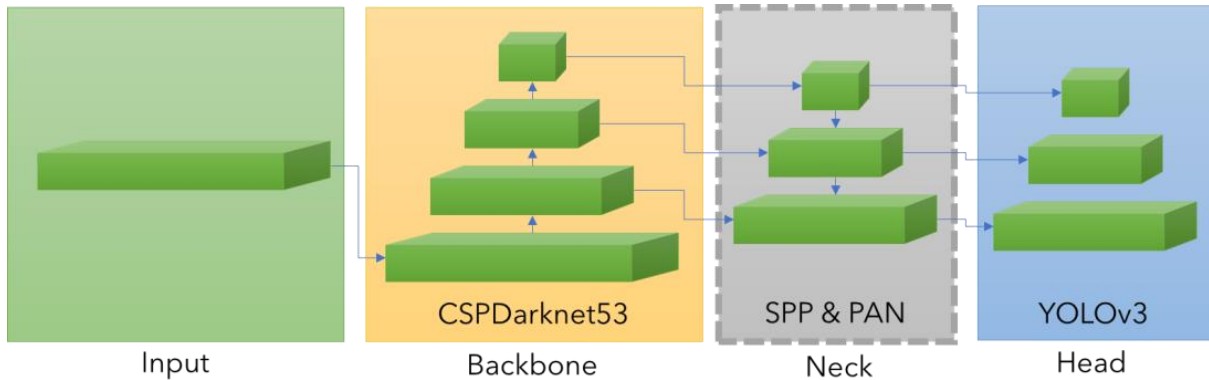

**Figure 5.** Simple architecture of YOLOv4 [44].

In this work, YOLOv4-Tiny is also used for study and comparison. YOLOv4-Tiny is a compressed and lite version of YOLOv4. The purpose of this compression is to reduce the computation so it can run on hardware with lower capacities, even for mobile or embedded devices. In this way, economic reasons can also be achieved. In addition, this algorithm has a higher speed but is less accurate than the full version (YOLOv4). In its research, YOLOv4-Tiny achieved 22.0% AP (average precision) or 42.0% AP50 at a speed of 443 FPS (frames per second), while YOLOv4 was able to achieve 43.5% AP or 65.7% AP50 at a real-time speed of 65 FPS for the MS COCO dataset. According to these results, YOLOv4-Tiny is approximately seven times faster than YOLOv4 but only has 2/3 of the accuracy. It makes YOLOv4-Tiny more suitable for cases that require high detection speed, unnecessarily high accuracy, applied to hardware with lower capacities (economic reasons), or for mobile or embedded devices [45]. YOLOv4-Tiny reduces the layers of some components of the original YOLOv4 to achieve a faster detection speed. First and foremost, the number of layers in the CSP backbone is reduced from 137 to only 29 pre-trained convolutional layers. In addition, YOLOv3 reduces the head from 3 to 2, and there are only a few anchor boxes for prediction [45].

These recognition algorithms were executed on CiRA-Core, a deep learning platform originally developed by Advanced Manufacturing Innovation (AMI) KMITL and first described in [46,47]. The advantages of this platform include its ease of use, user-friendliness, plethora of interface options, and provision of several automated processes, including during training, which will automatically select the most effective parameters for optimal results. In the training process, both YOLOv4 and YOLOv4-Tiny delivered good accuracy, between 0.15 and 0.03. The training was carried out with a batch size of 64 and 16 subdivisions, a learning rate of 0.001, and an adam optimizer. Data enrichment was carried out using rotation techniques, with a setting of 90 images per rotation (360°). The hardware used was a desktop PC with an Intel® 1151 Core™ i7-9700 3.0 GHz CPU (Central Processing Unit), NVIDIA GeForce RTX 3070 8 GB GDDR6 GPU (Graphical Processing Unit), and 32 GB DDR4/3200 RAM (Random Access Memory). Training time for each scheme took approximately 4–6 h.

*2.5. Validation Matrix*

The confusion matrix is used to evaluate the model's output in this work. This matrix is built from four blocks; TP, TN, FP, and FN. TP and TN are the basic truths, and FP and FN are the basic falsehoods. TP (True Positive) is defined as when the model can correctly detect the object (fish), TN (True Negative) is defined as when the model can correctly not detect the not-existent fish, which was not measured in this work, FP (False Positive) is defined as when the model incorrectly detects a fish, and FN (False Negative) is defined as when the model fails to detect the fish. From the confusion matrix, we can define the

accuracy to evaluate the model's output. It can be obtained from a comparison between the basic truth and the total blocks, as described by the following equation:

$$Accuracy = \frac{TP + TN}{TP + TN + FP + FN} \times 100\% \qquad (2)$$

where, *TP* = True Positives, *TN* = True Negatives, *FP* = False Positives, and *FN* = False Negatives.

In this work, the model's output is also categorized into three groups: correct detection, false detection (which includes wrong and double detection), and not-detect. Correct detection is defined as the detection and classification of the model that can be carried out correctly and consistently while the fish is fully visible in the frame. The wrong detection is determined if the model detects the wrong fish class more than once for more than 1 s. Double detection is specified if another fish class is also detected and appears more than once for more than 1 s. Not-detect is counted if the model cannot detect at all, can detect only momentarily (less than 1 s even though several times), or is unstable with a break of more than once for more than 1 s. The accuracy can be defined from a comparison between the number of correct detections and the total number of detections, as expressed in Equation (3). In addition to accuracy, several other parameters are also measured to evaluate the model by the confusion matrix, including precision, recall or sensitivity, specificity, and F-score. Those parameters are each obtained by Equations (4)–(7) [15].

$$Accuracy = \frac{\sum_i^N Pi}{\sum_i^N |Qi|} \times 100\% \qquad (3)$$

where, $\sum_i^N Pi$ is the number of correct detections, and $\sum_i^N |Qi|$ is the total number of all detections.

$$Precision = \frac{TP}{TP + FP} \times 100\% \qquad (4)$$

$$Recall/Sensitivity = \frac{TP}{TP + FN} \times 100\% \qquad (5)$$

$$Specivity = \frac{TN}{TN + FP} \times 100\% \qquad (6)$$

$$F - score = \frac{2 * Precission * Recall}{Precission + Recall} \times 100\% \qquad (7)$$

## 3. Experimental Results and Discussion

After the image data for training was prepared, the algorithm was trained with various image data input schemes, labeling techniques, and network versions to determine which approach was the most effective, including the proposed method. The trained algorithm was then applied to two high-speed videos, 1 and 2, for evaluation (also referred to as video tests after). The entire flow of this work can be seen in Figure 6, while the experimental results are summarized in Tables 5 and 6, Figures 7–9. The following is the explanation and discussion for every test result's scheme.

### 3.1. Using Static Pictures for Training Data

In this scheme, static images were used as training data in YOLOv4. The use of static images for this training is described in Section 2.2. Then, the algorithm that had been trained was tested on the video tests. From the experimental results, the output accuracy obtained was very low. In video test 1, the model could only correctly detect 2 of a total of 71 fish/detections. This means that the accuracy obtained is only 2.82%. In video test 2, the model could correctly detect 11 of 171 fish/detections, so the accuracy obtained is only 6.43%. The final average accuracy of this model (the average accuracy of video tests 1 and 2) is only 4.62%.

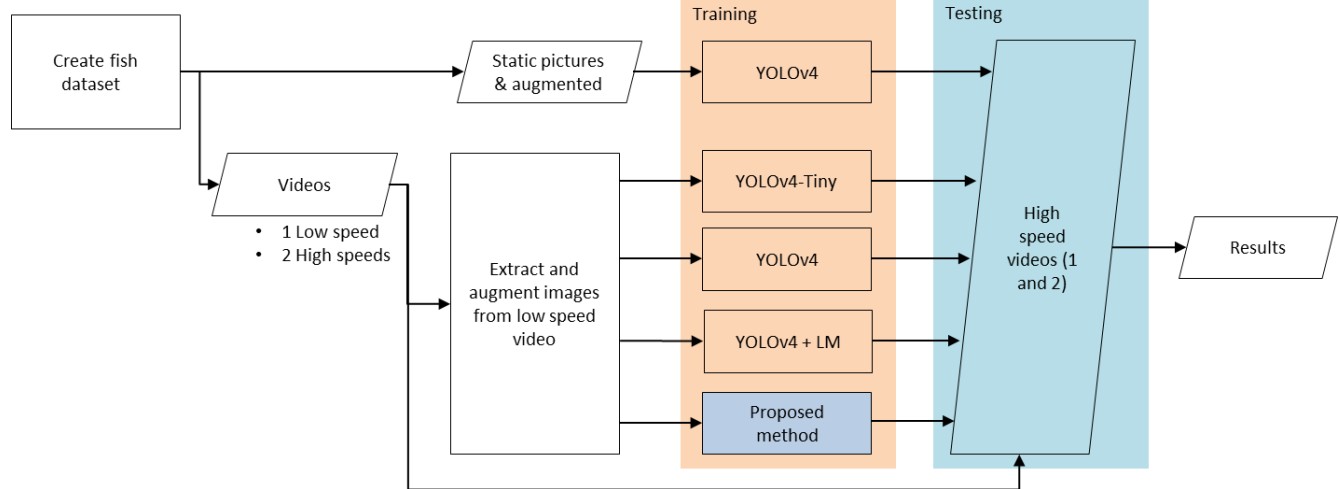

**Figure 6.** The work flowchart.

Certainly, the output accuracy result of this scheme is unacceptable. It indicates that using static images as training data for the algorithm, which is then used to recognize different images, is very ineffective. Although humans are able to classify fish species on static images, the learning process in the algorithm cannot capture the same thing. Static images have different sizes than the video tests used. The results of the accuracy and other performance parameters of this scheme can be seen in Figure 7.

### 3.2. With the Lite Version (YOLOv4-Tiny)

For the next step, the extracted pictures were used as training image data. Extracted pictures were obtained as described in Section 2.2. In this scheme, the lite version of YOLOv4 (YOLOv4-Tiny) was employed. This algorithm could be well trained and then tested on the same video tests. The output accuracy results obtained from the model were good enough. In video test 1, the model only detected four fish incorrectly out of 71 fish/detections. In video test 2, the model could correctly detect 154 out of 171 fish/detections while making 17 detection errors. This gives an accuracy score on video tests 1 and 2 of 94.37% and 90.06%, respectively, so the average score of both is 92.21%.

Of all the errors, the model made a double classification, i.e., one type of fish was detected as two different fish. It often appears between Yeesok and Nuanchan, Nuanchan and Tapian, and Jeen Ban and Jeen To. These fish are very similar to each other.

### 3.3. With YOLOv4 Using Conventional and Landmarking Labeling Techniques

At this stage, extracted pictures were used as training image data, and we used YOLOv4 as the algorithm. The first experiment was carried out using conventional labeling techniques. With this model, the accuracy output was better. In video test 1, an accuracy score of 97.18% was obtained, and in video test 2, it was 91.23%, so the average final accuracy of this model could reach 94.21%.

Although the final accuracy is only slightly better than with YOLOv4-Tiny, the total number of false (double) detections is much lower. From 4 to only 1 in video test 1, and from 17 to only 7 in video test 2. This means YOLOv4 has a higher classification accuracy capability than the lite version (YOLOv4-Tiny). However, another problem arose: the detection failure occurred nine times, all of which were in the Sawai class.

**Table 5.** Experimental results (accuracy).

| Approches | Video Test-1 | | | | | Video Test-2 | | | | | Average (Final Accuracy) (%) |
|---|---|---|---|---|---|---|---|---|---|---|---|
| | Correct Detection | False (Double/Wrong) Detection | Not Detect | Total Detection | Accuracy (%) | Correct Detection | False (Double/Wrong) Detection | Not Detect | Total Detection | Accuracy (%) | |
| YOLOv4 with Static Pics | 2 | 0 | 69 | 71 | 2.82 | 11 | 0 | 160 | 171 | 6.43 | 4.62 |
| YOLOv4-Tiny | 67 | 4 | 0 | 71 | 94.37 | 154 | 17 | 0 | 171 | 90.06 | 92.21 |
| YOLOv4 | 69 | 1 | 1 | 71 | 97.18 | 156 | 7 | 8 | 171 | 91.23 | 94.21 |
| YOLOv4 + LM | 68 | 6 | 0 | 74 | 91.89 | 159 | 11 | 1 | 171 | 92.98 | 92.44 |
| Proposed method | 72 | 0 | 1 | 73 | 98.63 | 167 | 4 | 0 | 171 | 97.66 | 98.15 |

**Table 6.** Experimental results (other performance parameters).

| Approches | Video Test 1 and 2 | | | | | | |
|---|---|---|---|---|---|---|---|
| | Correct Detection (TP) | Wrong/Double Detection (FP) | Not Detect (FN) | Total Detection | Precision (%) | Sensitivity (%) | F-Score (%) |
| YOLOv4 With Static Pics | 13 | 0 | 229 | 242 | 100.00 | 5.37 | 10.20 |
| YOLOv4-Tiny | 221 | 21 | 0 | 242 | 91.32 | 100.0 | 95.46 |
| YOLOv4 | 225 | 8 | 9 | 242 | 96.57 | 96.15 | 96.36 |
| YOLOv4 + LM | 227 | 17 | 1 | 245 | 93.03 | 99.56 | 96.19 |
| Proposed Method | 239 | 4 | 1 | 244 | 98.35 | 99.58 | 98.96 |

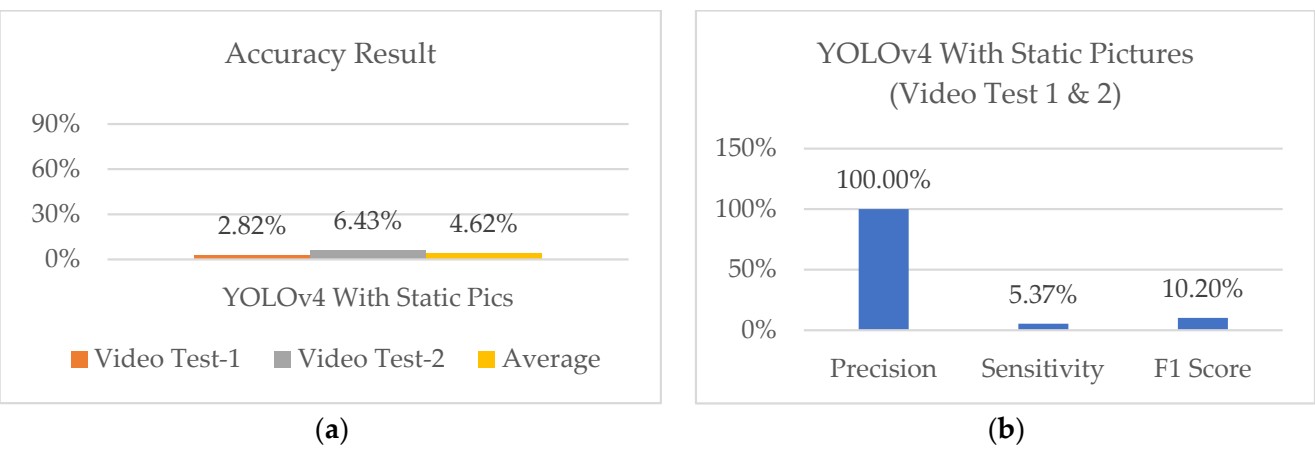

**Figure 7.** Test results of YOLOv4 with static pitures: (**a**) accuracy; (**b**) other parameters.

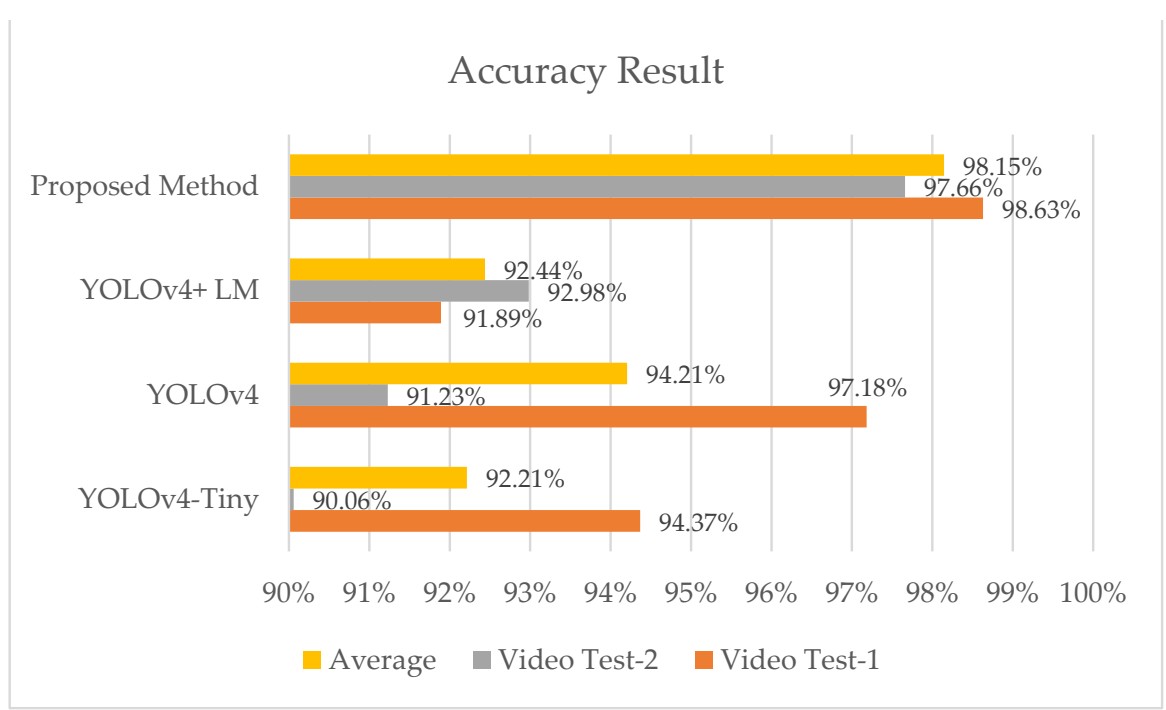

**Figure 8.** Experimental results (accuracy).

In the labeling process for the next scheme, YOLOv4 will be combined with the landmarking technique. With this approach, the results obtained were 91.89% accuracy for video test 1 and 92.98% accuracy for video test 2, so the average final accuracy was 92.44%. This model did many double detections, especially for the Nuanchan and Tapian, which have the same appearance of scales but different shapes. Many double detections were also carried out for the Jeen Ban-Jeen To classes. In addition, the model also detected scattered fish outside the conveyor incorrectly. The three scattered Yeesok fish were detected as Nin.

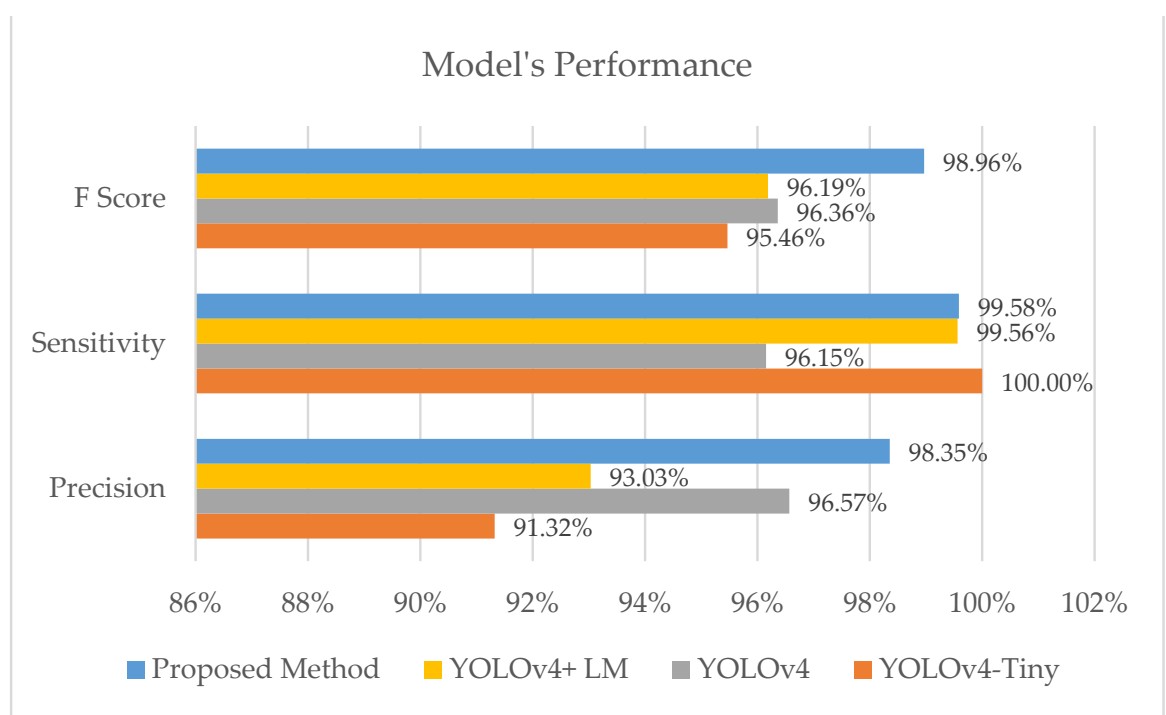

**Figure 9.** Experimental results (other performance parameters).

*3.4. With the Prophosed Approach*

Up to this step, some of the findings from various trials can be highlighted:

1.  Using extracted pictures as training data provided much more effective results than static pictures.
2.  YOLOv4 provided better accuracy results than its lite version (YOLOv4-Tiny).
3.  Using conventional labeling techniques on YOLOv4 gave fairly accurate detection results, even for fish classes that were similar, but many of them failed to detect the Sawai class.
4.  Combining YOLOv4 with the landmark labeling technique provided a fairly accurate detection result. Still, it generated many double detections for similar classes of fish, mostly for Nuanchan and Tapian, as well as Jeen Ban and Jeen To.

We can focus on the YOLOv4 algorithm with extracted pictures as image training data, and this approach produced the best accuracy. Then, we focus on the labeling technique used. The Sawai class was detected poorly using YOLOv4 and conventional labeling technique. If observed, this type of fish is the biggest. When it appears on the video, the size of this fish almost fills the entire frame. So that when extracted images of this Sawai fish are generated and used as a training image, utilizing conventional labeling techniques, many other objects in the fish's background will be included in the training process. These objects are not constant (not the same in every extracted picture), making the YOLOv4 algorithm less effective in capturing Sawai fish features because it mixes with other objects in the background [38].

In contrast to other smaller fish (seven other classes) such as Nin, Nuanchan, Tapian, and even Jeen Ban and Jeen To, even though the background is involved in the training process, the background tends to be constant (almost just a conveyor background). This condition, on the other hand, has a good effect on the learning process. The model only detects fish on the conveyor, so it does not detect fish scattered outside it. The model can better extract the shape features of the fish so that it can better distinguish between Nuanchan and Tapian fish, which have a similar appearance of scales, tails, and heads but can be distinguished by their shape.

YOLOv4 combined with the landmark labeling technique, resulted in lower classifiability for similar fish such as Nuanchan-Tapian and Jeen Ban-Jeen To. Because, as previously described, with this technique, the background of the fish is completely removed so that the algorithm is not affected by the background during the training process. The advantage of this approach is that the model can better detect Sawai fish because the algorithm is not affected by the background object, which is inconsistent. However, this makes the algorithm not good at extracting shape features. Hence, the model experiences a lot of false (double) detection for fish that should be able to be distinguished from their shapes, such as Nuanchan-Tapian and Jeen Ban-Jeen To. In addition, the model also detects fish that are outside the conveyor (scattered fish).

This hypothesis can be supported by visualizing the feature maps—the features captured by the algorithm during the learning process. The visualization of these feature maps can be seen in Figure 10 for the algorithm using the conventional labeling technique and in Figure 11 when using landmarking. The example taken is the same fish for easy comparison. Figure 10 shows that the background is included in the training process and extracted by the algorithm (see the first convolution process (conv2d)). Even the background is still carried over and becomes part of the extracted features in a deeper layer; pooling 1 (max_pooling2d), convolution 2 (conv2d_1), pooling 2 (max_pooling2d_1), and convolution 3 (conv2d_2). This means that the background also becomes one of the determinants or features that are considered when the algorithm runs to detect the fish. In addition, the features of the fish's shape are better because there is a background comparison.

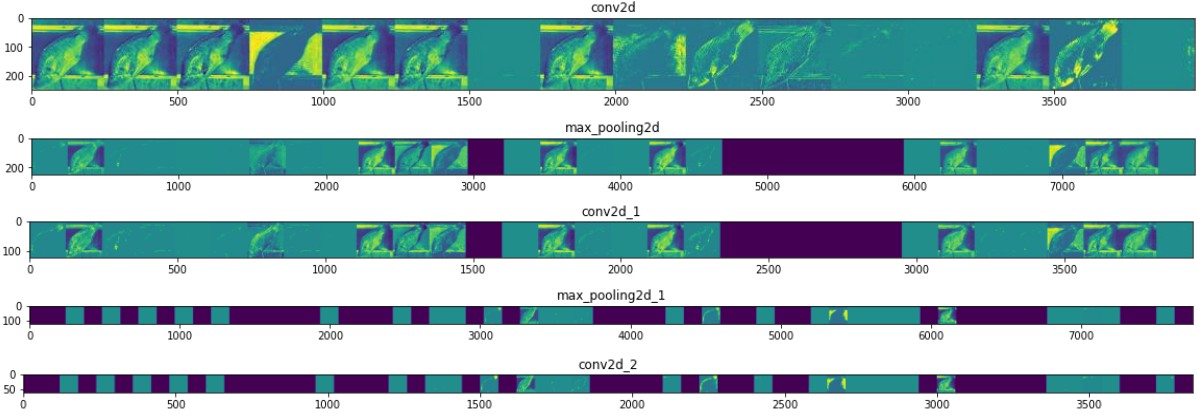

**Figure 10.** Feature maps visualization with conventional labeling.

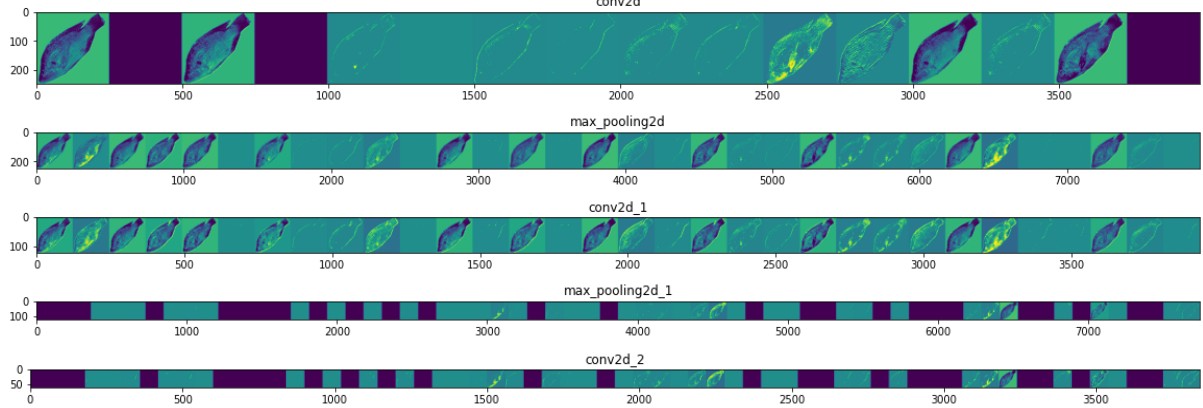

**Figure 11.** Feature maps visualization with landmarking labeling.

On the other hand, the extraction process only happens on fish objects on feature maps made from the learning process on YOLOv4 and the landmark labeling technique. This is because the background of the fish is completely removed with this technique. It can be seen in Figure 11. Because the learning process only focuses on the fish object, it is not disturbed by other objects. However, the extraction of shape features is less good because there is no background for comparison. For that, an approach is proposed. This approach uses extracted images as training data, YOLOv4, and a combination of labeling techniques. The combined labeling technique uses the landmarking technique for large fish (Sawai) and conventional for other classes. Then this approach is tested on the same video tests.

From the test results, the accuracy obtained was 98.63% for video test 1 and 97.66% for video test 2. This means that the average final accuracy was 98.15%. This achievement increased significantly compared to using the previous methods. There were only 4 false (double) detections, which had decreased a lot from the YOLOv4 with the landmark labeling technique (17 detections) and the conventional (8 detections). In addition, the detection failure was also significantly reduced to only one from the previous nine (using YOLOv4 with conventional labeling techniques). In other words, this approach can reach its optimum point by combining existing labeling techniques and avoiding each of their weaknesses.

### 3.5. Comparison with Recent State of Art

The method proposed in this work is compared to the current state of the art in order to identify its most significant contributions. Recent years have seen the development of at least six studies on recognizing swimming fish, which were also discussed in the Section 1. Table 7 provides a comparison summary. The table demonstrates that the proposed Method with simple and effective algorithms has the best average recognition results and has been tested on aquaculture fish video datasets; therefore, this work is most applicable to fish sorting systems in the fish industry.

### 3.6. Limitations and Future Developments

Limitations of this work also need to be reported. The biggest limitation is that the algorithm is standard and cannot be modified with the tools used in this work. So optimization studies cannot be carried out within the scope of modifications to the YOLOv4 or YOLOv4-Tiny algorithms. For this reason, this limitation can also be developed in the future. Optimization studies can be implemented by modifying them to achieve more optimal accuracy output or more efficient computations by the other tools. Or the use of a higher version (YOLO versions 5, 6, or 7) can also be considered. In addition, future developments can also be conducted using image processing techniques such as reducing glare on the appearance of fish objects or combining approaches with other classification algorithms or unsupervised learning. A tuning or inference method can be proposed to optimize the accuracy of the approach in this work. Moreover, a statistical hypothesis analysis could be conducted.

**Table 7.** Comparison chart of previously related works with the proposed method.

| Related Work | Fish Dataset | Fish Type | No. of Fish Classes | Method/ Algorithm | Accuracy (%) | Precision (%) | Sensitivity (%) | F Score (%) | Advantage | Disadvantage |
|---|---|---|---|---|---|---|---|---|---|---|
| [7] | own dataset | deep ocean fish | 20 | CNN | 94.90 | - | - | - | (1) The best accuracy is achieved by recognizing nine species of fish. (2) Recognition results are accurate even though the backgrounds are various or the fish only partially appear. | (1) The results of accuracy can still be increased. (2) High accuracy is expected to apply to all classes. |
| [8] | own dataset | deep ocean fish | 1 | Multi-cascade object detection Network, 7 CNNs 2 RPNs, trained LSTMs | - | 67.28 | 68.25 | 67.76 | Promising to detect and count fish under various benthic backgrounds and illumination conditions. | Only detect fish, not classify them. |
| [9] | Fish4-Knowledge & UWA | deep ocean fish | 17 | Optical flow, GMM, ResNet-50, YOLOv3 | 91.64 | - | - | 95.47 | Quite effective, even applied to many classes with diverse backgrounds and illumination challenges. | The results of accuracy can still be improved |
| [16] | own dataset | Aquacultured fish | 1 | Image enhancement, YOLOv3 | 100 | - | - | - | (1) Effectively detect all fish in the test images. (2) Image Enhancement can optimize the work of the algorithm significantly. | Only to detect fish and trajectory, not for classification. |
| [17] | own dataset | Aquacultured fish | 1 | Faster R-CNN, YOLOv3 | 98.13 | - | - | - | (1) High accuracy is obtained from Faster R-CNN. (2) Simple with good results | Only to detect fish and trajectory, not for classification. |
| [30] | Fish4-Knowledge | deep ocean fish | 1 | GMM, Pixel-wise posteriors, CNN | - | - | - | 87.44 | Increased the result fairly from the previous work. | (1) Only detect fish, not classify them. (2) The result can still be improved |
| Proposed method | own dataset | Aquacultured fish | 8 | Optimized YOLOv4 | 98.15 | 98.35 | 99.58 | 98.96 | (1) Simple method but delivers high results. (2) Ready to implement for aquaculture fish sorting system. | (1) Using not open access deep learning software. (2) YOLOv4 can not be modified. |

-: not reported.

## 4. Conclusions

This work aims to propose an optimal approach for detecting and classifying fish intended for the aquaculture industry, especially for making automatic sorting processes. In this work, we created and presented a real video dataset of freshwater fish running on a conveyor, which is the first and only, as far as the authors know. The dataset includes eight types of freshwater fish that are grown and eaten most often in Thailand and nearby countries. Some of these fish are native to Thailand. This work uses YOLOv4, the most viral algorithm for object detection, and a relatively new version. Several studies were conducted to determine the level of accuracy, and finally, an approach was proposed.

This approach utilizes YOLOv4, optimized with a combination/custom labeling technique, and extracted images as training data. From the test results on the video of eight types of freshwater fish running on a conveyor with a total duration of 25 min and 37 s at a speed of 505.08 m/h, the model could produce an accuracy of 98.15%. These results are considered quite good and can even be improved in the future. By using real videos of freshwater fish running on a conveyor, this work is expected to contribute to the development of fish detection and classification, especially for the automatic sorting process in the fish industry, which is very close to real conditions.

**Author Contributions:** Conceptualization, A.K. and T.S.; methodology, W.T. and N.N.; software, A.K. and G.S.; validation, W.T. and G.S.; formal analysis, A.K. and N.N.; investigation, W.T. and G.S.; resources, T.S. and N.N.; data curation, A.K. and T.S.; writing—original draft preparation, A.K.; writing—review and editing, A.K., T.S., W.T., N.N. and G.S.; visualization, A.K. and N.N.; supervision, W.T. and G.S.; project administration, W.T.; funding acquisition, T.S. All authors have read and agreed to the published version of the manuscript.

**Funding:** King Mongkut's Institute of Technology Ladkrabang Research Fund: KREF206314.

**Institutional Review Board Statement:** Not applicable.

**Informed Consent Statement:** Not applicable.

**Data Availability Statement:** The dataset created and the results videos can be found at: https://drive.google.com/drive/folders/1OrEUIYdiDWbvDyUV46WSrr6Yd83OTuwt?usp=sharing (accessed on 1 March 2023). The python codes and repository are shared at: https://github.com/AriKus77/Fish-Recognition-for-Auto-Sorting--Feature-Extraction (accessed on 1 March 2023).

**Acknowledgments:** We want to thank AMI (Advanced Manufacturing Innovation)-KMITL, Bangkok, Thailand, for granting the CiRA-Core software license, and declare that this work was fully supported by King Mongkut's Institute of Technology Ladkrabang (KMITL).

**Conflicts of Interest:** The authors declare no conflict of interest.

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
