# Peer review of "Fish Detection and Classification for Automatic Sorting System with an Optimized YOLO Algorithm"

_applsci, doi:10.3390/app13063812_

Round 1

Reviewer 1 Report

- " [7] employed CNN 52 (Convolutional Neural Network) to classify fish by training ..." a sentence should not start with a reference. Pls update this through out the article.

- Discuss about the key contributions of this study.

- Summarize the recent works in teh form of a table.

- Some of the recent works on iout such as the following can be discussed "

Federated Learning for IoUT: Concepts, Applications, Challenges and Future Directions, Blockchain for Internet of Underwater Things: State-of-the-Art, Applications, Challenges, and Future Directions"

- Compare the results obtained with recent state of the art.

- What is the computational complexity of the proposed approach?

- The authors can enhance the results analysis by performing a statistical hypothesis analysis.

Reviewer 2 Report

This paper proposes an approach based on YOLOv4 and optimized with a unique labelling technique to detect and classify fish in real videos of 8 species of aquacultured freshwater fish moving on a conveyor. The research design is appropriate, and research results and findings are clearly presented. There are some suggestions for the further revision of the paper draft.

1. The introduction is well structured and shows a clear need for research work. Related works about the detection and classification of fishes are reviewed. To better present the advantages of CNN (convolutional neural networks) in this area, more studies on the application of CNN for classification and detection should be introduced, such as: doi:10.3390/app12094356; doi:10.1007/s00170-022-10335-8

2. Section 2.4 lists the training setting of the proposed CNN model, including batch size, learning rate, optimizer, etc. Did the authors test the training performance of other parameters? Or the authors can clarify why this specific combination of training parameters is adopted.

3. This paper compares the performance of the proposed method with other YOLO models. It is recommended to conduct a comprehensive comparison study and other state-of-the-art CNN networks should be covered.

4. The conclusion is well structured and shows the value of the work clearly. What the authors’ plan for the relevant codes associated to this work? Will the code be publicly available in the future?

Round 2

Reviewer 1 Report

All the comments are addressed